# Durability and Performance of Encapsulant Films for Bifacial Heterojunction Photovoltaic Modules

**DOI:** 10.3390/polym14051052

**Published:** 2022-03-06

**Authors:** Marilena Baiamonte, Claudio Colletti, Antonino Ragonesi, Cosimo Gerardi, Nadka Tz. Dintcheva

**Affiliations:** 1Dipartimento di Ingegneria, Università di Palermo, Viale delle Scienze, Ed. 6, 90128 Palermo, Italy; marilena.baiamonte@unipa.it; 23SUN-Enel Green Power SpA Contrada Blocco Torrazze, Zona Industriale Catania, 95121 Catania, Italy; claudio.colletti@enel.com (C.C.); antonino.ragonesi@enel.com (A.R.); cosimo.geraradi@enel.com (C.G.)

**Keywords:** encapsulants for PV modules, lamination process, EVA, POE

## Abstract

Energy recovery from renewable sources is a very attractive, and sometimes, challenging issue. To recover solar energy, the production of photovoltaic (PV) modules becomes a prosperous industrial certainty. An important material in PV modules production and correct functioning is the encapsulant material and it must have a good performance and durability. In this work, accurate characterizations of performance and durability, in terms of photo- and thermo-oxidation resistance, of encapsulants based on PolyEthylene Vinyl Acetate (EVA) and PolyOlefin Elastomer (POE), containing appropriate additives, before (pre-) and after (post-) lamination process have been carried out. To simulate industrial lamination processing conditions, both EVApre-lam and POEpre-lam sheets have been subjected to prolonged thermal treatment upon high pressure. To carry out an accurate characterization, differential scanning calorimetry, rheological and mechanical analysis, FTIR and UV-visible spectroscopy analyses have been performed on pre- and post-laminated EVA and POE. The durability, in terms of photo- and thermo-oxidation resistance, of pre-laminated and post-laminated EVA and POE sheets has been evaluated upon UVB exposure and prolonged thermal treatment, and the progress of degradation has been monitored by spectroscopy analysis. All obtained results agree that the lamination process has a beneficial effect on 3D-structuration of both EVA and POE sheets, and after lamination, the POE shows enhanced rigidity and appropriate ductility. Finally, although both EVA and POE can be considered good candidates as encapsulants for bifacial PV modules, it seems that the POE sheets show a better resistance to oxidation than the EVA sheets.

## 1. Introduction

Today, energy recovery from renewable sources and processes with less environmental and human health impacts, and the gradual release of traditional fossil fuel sources, due to their high carbon dioxide and pollutants production, are challenges and necessary issues. Therefore, the energy recovery considering sunlight, winds and tides is extremely attractive and specifically, the development of solar photovoltaic (PV) devices for efficient energy recovery is one of the most important research fields. As documented, the energy demand increases continuously, and is expected to reach around 778 exajoules (EJ) by 2035 [1].

Currently, 3SUN-ENEL Green Power (Catania, Italy) develops a new innovative device for efficient energy recovery, and particularly, in Figure 1a,b, a high reliable bifacial glass–glass heterojunction PV module is shown. This innovative heterojunction technology, combining amorphous and crystalline silicon, offers high performance and efficiency in energy recovery, even in extreme climatic conditions [2]. An important issue in PV modules construction and assembling is the use of appropriate encapsulant materials that can protect efficiently the active PV elements ensuring device high performance and durability [3,4]. The encapsulant polymer-based materials must protect PV modules efficiently against humidity, oxygen and other gas, must be transparent and flexible and must have a good adhesion with glass and solar cells [5,6,7,8]. Different encapsulant materials, such as polydimethylsiloxane (PDMS), poly-ethylene vinyl acetate (EVA), polyvinyl butyral (PVB), thermoplastic polyolefins (TPO), polyolefin elastomer (POE), have been considered suitable for industrial purpose [9,10,11,12,13].

Considering the balance between costs and performance, the best polymer material as PV encapsulant is EVA, and furtherly to improve its environment resistance, the EVA is added with crosslinking agents and appropriate stabilizers [13]. Therefore, EVA degrades upon solar exposure, even if using crosslinking agents and stabilizers [14,15,16]. As documented, the EVA degradation proceeds with acetic acid development and the latter leads to encapsulants yellowing, compromising the PV module function [17,18,19].

However, 3SUN researchers and partners, related to compatibility assessment of commercial EVA, POE, TPO and Ionomer films as encapsulants for bifacial heterojunction PV modules, highlight that the polyolefin elastomers are more compatible to heterojunction technology than other considered commercial materials. Assembling full-size (72 cells) modules, no failures induced by the POE encapsulant are observed after 3000 h in damp heat conditions, 600 thermal cycles and a sequential test using 60 kWh/m^2^ exposure [2].

Therefore, the published paper by Baiamonte et al. [20] proposes the formulation of encapsulants for bifacial heterojunction PV modules based on blends containing poly-ethylene vinyl acetate and polyolefin, i.e., EVA/PO, crosslinking agent and stabilizers, such as UV-adsorber, anti-oxidant and metal deactivator. All obtained results suggest that EVA/PO = 75/25 wt/wt%, containing crosslinking agent and stabilizers, show better mechanical behavior, optical properties and durability than that of neat EVA, suggesting a beneficial effect of the polyolefin presence at low amount. Besides, the photoxidation resistance of EVA/PO = 75/25 wt/wt% blend containing crosslinking agent and stabilizers is very similar to that experienced by neat EVA, highlighting that this blend is a good candidate as encapsulant material for bifacial PV modules.

In this work, the properties and performance of commercial EVA and POE sheets, before (pre-) and after (post-) lamination, as appropriate encapsulant materials for bifacial heterojunction PV modules are investigated. Accurate calorimetric, rheological and durability analysis, in terms of photo- and thermo- oxidation resistance, of both pre- and post- laminated EVA and POE are carried out, also considering the ability of these materials in heterojunction technology for PV modules assembling. However, commercial EVA and POE films are subjected to accelerated UVB exposure and prolonged thermal treatment, and their oxidation resistance is monitored by spectroscopic analysis in time.

Therefore, this proposed comparative study suggests and encourages further research studies regarding the formulation and discovery of PV encapsulants, with good performance, in terms of durability and oxidative resistance, and relatively low cost.

## 2. Materials and Methods

### 2.1. Materials

Commercial PolyEthylene Vinyl Acetate (EVA) and PolyOlefin Elastomer (POE) sheets, suitable for low UV cutoff, are purchased by Specialized Technology Resources Inc. Both EVA and POE contain appropriate crosslinking agent and stabilizing additives, such as antioxidants and hindered light amine stabilizers, as produced by manufacture. All additives have been added to EVA and POE during sheets formulation by producer. There are four different sheets considered: EVA pre-laminated (EVApre-lam), EVA post-laminated (EVApost-lam) and POE pre-laminated (POEpre-lam), POE post-laminated (POEpost-lam). The thicknesses of pre-laminated sheets are about 450 μm and to simulate industrially viable lamination process, the pre-laminated EVA and POE sheets have been subjected to pressure at 1 atm and temperature at 150 °C up to 20 min.

### 2.2. Characterizations

Differential Scanning Calorimetry: The calorimetric data were evaluated by differential scanning calorimetry (DSC) using a DSC60-Shimadzu calorimeter. All experiments were performed under dry nitrogen on samples of about 10 mg in 40 μL sealed aluminum pans. For both EVA and POE, the calorimetric scans, heating: from −80 to 120 °C and cooling: from 120 to −80 °C, were performed for each sample at scanning heating/cooling rate of 10 °C/min. The values of heat flow have been normalized considering sample mass.Rheological analysis: Rheological tests were performed using a stress-controlled rheometer (Rheometric Scientific, SR5, mod. ARES G2 by TA Instrument, New Castle, DE, USA) in parallel plate geometry (plate diameter 25 mm). The complex viscosity (η*), storage (G’) and loss (G”) moduli were measured under frequency scans from ω = 10 − 1 to 102 rad/s at T = 140 °C and T = 170 °C for EVA and POE, respectively. The strain amplitude was γ = 5%, which preliminary strain sweep experiments proved to be low enough to be in the linear viscoelastic regime.FTIR Spectroscopy: A Fourier Transform Infrared Spectrometer (Spectrum One, Perkin Elmer) was used to record IR spectra using 16 scans at a resolution of 1 cm^−1^. ATR-FTIR for some surface analysis has been also carried out, using 16 scans at a resolution of 1 cm^−1^. The progress of both photo- and thermo-oxidation degradation for EVA and POE has been followed by running FTIR analysis with time and monitoring the variations in the hydroxyl range (3200–3600 cm^−1^) and carbonyl range (1800–1500 cm^−1^) in time, using Spectrum One software.UV-visible Spectrometer, (Specord^®^250 Plus, Analytikjena, Torre Boldone, Italy), was used to record UV-Vis spectra performing 8 scans between 200 and 1100 nm at a resolution of 1 nm. The values of linear attenuation coefficient (k) were calculated considering the measured absorption values (A) and sample thickness (D), using the formula: k = A/(2.3D).

### 2.3. Accelerated Weathering and Thermo-Oxidation

Photoxidation was carried out using a Q-UV/basic weatherometer (from Q-LAB, Westlake, OH, USA) equipped with UVB lamps (313 nm). The weathering conditions were a continuous light irradiation at T = 70 °C.

Thermo-oxidation was carried out in a ventilated oven at 70 °C a time up to ca. 3500 h for both EVA and POE post-laminated sheets.

The progress of both photo- and thermo-oxidative degradation was followed by FTIR spectroscopic technique.

## 3. Results

### 3.1. Differential Scanning Calorimetry (DSC) Characterization

The identification of the transition temperatures for both commercial EVA and POE sheets is performed through differential scanning calorimetry. In Figure 2a,b, the thermograms from −80 °C up to 120 °C of both pre-laminated and post-laminated EVA and POE materials are plotted, and in Table 1, the main identified transition temperatures are reported. In Figure 2a, the glass transition between −40 °C and −20 °C, i.e., Tg around −36 °C, for both EVApre-lam and EVApost-lam samples is detectable and this transition is well noticeable for post-laminated sample. It can be observed that EVApre-lam shows three endothermic peaks in the range from +30 °C up to +90 °C, see blue curve in Figure 2a. The first small peak at about +30 °C, probably, can be attributed to the presence of low molecular weight additives, with low temperature fusion transition. The other two fusion peaks, at about +55 °C and +85 °C, respectively, can be attributed to the fusion transition of two different crystalline structures of EVA sample. After the lamination, the thermogram of EVApost-lam appears slightly different, see red curve in Figure 2a, and there are two noticeable small exothermic peaks in the range +10 °C up to +35 °C, probably, due to the occurrence of crosslinking and additives dispersion during lamination upon prolonged thermal treatment at high pressure. Interestingly, the peak at about +30 °C is not well distinguished and a very broad shoulder in the range between 30 and 50 °C can be observed, highlighting a structural change in the organization of the low molecular weight additives and their interaction with EVA macromolecules. Besides, both fusion peaks at about +55 °C and +85 °C become well pronounced, pointing out the presence of two different polymer crystalline structures. Surprisingly, the fusion enthalpy for EVA increases ca. 1.6 times upon lamination process, suggesting the formation of better ordered 3D-structures, see last column in Table 1.

In Figure 2b, the thermogram of POEpre-lam and POEpost-lam samples are plotted. Moreover, in this case, both POEpre-lam and POEpost-lam samples show a glass transition at around −25 °C and no significant different for glass transition of these samples is observed before and after lamination process. The POEpre-lam sample shows two well visible fusion peaks in the temperature range from +50 up to +100 °C, see blue curve in Figure 2b. It can be observed that after the lamination process both peaks at about +60 °C and +95 °C become well pronounced, pointing out again the presence of two different polymer crystalline structures. Interestingly, the increase of fusion enthalpy for POE upon lamination is ca. 2.7, suggesting the formation of better ordered 3D-structure also for POE, see last column in Table 1.

To sum up, it is worth noting that the glass transition and exothermic phenomena for both EVA and POE are almost uninfluenced by lamination process, while the fusion occurrence reveals that the lamination process could be considered responsible for the formation of a large amount of 3D-ordered crystalline structures. Specifically, the total peaks areas of EVApost-lam (from +25 °C up to 95 °C) and POEpost-lam (from +30 up to +110 °C) samples are higher ca. 1.6 times and 2.7 times than the peak areas of EVApre-lam and POEpre-lam samples, respectively. Based on these results, it can be supposed that the lamination process has a beneficial effect on the formation of 3D-ordered crystalline structures, and it seems that the final POE structure is better structured than the EVA one.

### 3.2. Rheological Characterization

In Figure 3, the trends of storage and loss moduli, G’ and G”, and complex viscosity, η*, as a function of the frequency of both pre-laminated and post-laminated EVA and POE materials are plotted. The rheological behavior of EVApre-lam and EVApost-lam are slightly different and it is worth noting that for both EVApre-lam and EVApost-lam, no Newtonian plateau is observed, and well pronounced shear thinning is visible, suggesting the presence of crosslinked 3D-structure. Unexpectedly, the values of both moduli G’ and G” and complex viscosity are lower than the values of before the lamination process, and the latter could be understand considering that during prolonged lamination process, i.e., up to 20 min at high temperature and pressure, the EVA underwent thermal degradation, which leads to the formation of volatile acetic acid.

Therefore, the elimination of acetic acid molecules during lamination causes decrease for both moduli and viscosity, although crosslinking also occurs. The latter is understandable considering that in the melt state, the macromolecules of EVApost-lam are able to move themself and the system rigidity is lower than EVApre-lam. Interestingly, the G’ and G” trends remain almost parallel between them, i.e., no cross-over point is observed, for both EVApre-lam and EVApost-lam, suggesting the presence of crosslinked structure, which does not change significantly upon lamination.

Contrarily, the viscosity of POEpost-lam is significantly higher than the viscosity of POEpre-lam, i.e., the difference is more than one decade, and additionally, the slopes of trends are different, highlighting a beneficial effect of the lamination process on POE crosslinking. The change from solid-like to liquid-like behavior for pre-laminated and post-laminated POE occurs at different frequencies, i.e., the cross-over point changes from 1.58 rad/s for POEpre-lam to 25.11 rad/s for POEpost-lam. Therefore, the rheological behavior of pre-laminated POE sample reveals the existence of no well-3D-structured sample, and in this case also, no Newtonian plateau is noticed. The rheological behavior of POEpost-lam is significantly changed upon lamination process and there is a well-3D-structured crosslinked sample.

Based on the rheological behavior, it can be surmised that the lamination process has a well pronounced beneficial effect on 3D-structuration for POE, rather than for EVA. After the lamination, the EVA sample exhibits an affination of existing 3D-structure, without significant change in the melt state behavior. The POEpost-lam shows solid- to liquid-like transition at high frequency, a significant viscosity enhancement and well pronounced shear thinning in comparison to POEpre-lam, highlighting a very good 3D-structuration.

### 3.3. Mechanical Characterization

In Figure 4, typical stress-strain curves of pre-laminated and post-laminated EVA and POE samples are plotted, and in Table 2, obtained values of main mechanical properties, i.e., elastic modulus, E, tensile strength, TS, and elongation at break, EB, are reported. It is clearly noticeable that the lamination process has a positive effect on the rigidity of both EVA and POE, i.e., the values of elastic modulus after the lamination increase about 45% more than the values before lamination.

As expected, for EVA sample, upon lamination, the tensile strength increases about 70%, while the elongation at break is reduced about 40%. Interestingly, for POE sample, upon lamination, the tensile strength increases about 48%, while the elongation at break remains almost unchanged. These results are understandable considering that during the lamination, the crosslinking process occurs, and this leads to an increase of rigidity, also according to the results by calorimetry and rheological analyses, above commented.

### 3.4. UV-Visible Characterization

In Figure 5a,b, the linear attenuation coefficient (K) of both pre- and post- laminated EVA and POE are plotted, respectively. The values of linear attenuation coefficient for all samples are calculated using the formula reported in the experimental part, i.e., considering the absorption values (A) and sample thicknesses (D). As known, the material is almost transparent when K value is close to zero. It is clearly noticeable that the EVApost-lam and POEpost-lam samples show K values lower than the EVApre-lam and POEpre-lam samples, especially in the visible range, although the thicknesses of both post-laminated samples are two times higher than the pre-laminated counterparts. This behavior is due to the lamination process having a beneficial effect on both occurrence of 3D-structuration for both EVA and POE and additives dispersion and distribution. Additionally, the small shoulders at about 290 nm in all K trends can be attributed to the presence of stabilizing molecules, and their presence is clearly noticeable before and after lamination.

### 3.5. FTIR Characterization

In Figure 6a,b, the FTIR spectra of both pre- and post- laminated EVA and POE are plotted, respectively. It is worth noting that the main absorption bands (ca. 2800–2900 cm^−1^, due to CH stretching, ca. 1700 cm^−1^ due to carbonyl band stretching, and other bands in 1400–800 cm^−1^, due to different chemical nature and structures) in FTIR spectra are saturated because there are thick original commercial samples. According to literature, the main representative FTIR ranges for polyolefins and polyolefin derivates are both carbonyl (ca. 1600–1800 cm^−1^) and hydroxyl (3200–3600 cm^−1^) range, and additionally, the oxidation degradation of these polymers can be profitable following the monitoring of changes in these two main ranges. It is worth noting that in the spectra of EVApre-lam a small shoulder at ca. 1650 cm^−1^ is noticeable and this could be attributed to the presence of some unsaturation in this material. In the spectra of EVApost-lam, the shoulder at ca. 1650 cm^−1^ is not visible, also because the carbonyl bands appear larger due to higher sample thickness, while a small shoulder at ca. 1780 cm^−1^ appears, probably, due to the formation of some esters during prolonged lamination process. Besides, the hydroxyl bands in EVApost-lam spectra appear more pronounced than the bands in the spectra of EVApre-lam. Similar consideration can be made also for the spectra of POEpre-lam and POEpost-lam samples; upon the lamination process, in the spectra of POEpost-lam a small shoulder at ca. 1650 cm^−1^ appears and the hydroxyl bands are more pronounced in comparison to POEpre-lam.

### 3.6. Photoxidation Resistance

To investigate the photoxidation resistance of EVA and POE, the original sheets have been subjected to UVB exposure and the degradation has been monitored by FTIR analysis in time. In Figure 7a–d, the FTIR of EVApre-lam, EVApost-lam, POEpre-lam and POEpost-lam commercial samples at different exposure times are plotted.

Therefore, according to the literature, EVA photodegradation proceeds with accumulation of oxidation products leading to the formation of new absorption bands in the carbonyl domain (shoulders at 1780 cm^−1^ and 1715 cm^−1^ in IR spectra), in the hydroxyl domain (3200–3600 cm^−1^) and acetic acid formation, which leads to pH lowering and corrosion ability increasing. Moreover, EVA shows a very fast yellowing, due to the formation of oxidation products, and to avoid unwanted effects, the addition of stabilizers is imperative, especially for manufacturing in service upon sunlight [21,22]. However, as well known, the photodegradation of polyolefins and polyolefins-based polymers proceeds mainly with accumulation of groups in carbonyl domain (1600–1800 cm^−1^) and hydroxyl domain (3200–3600 cm^−1^), and subsequently, worsening of their macroscopic properties [20,23,24,25]. Considering all these issues and FTIR analysis of these commercial EVA and POE samples, reported above, the progress of photoxidation for both EVA and POE can be followed profitable accounting the changes in carbonyl and hydroxyl domains. Besides, as commend above, on the FTIR spectra, main absorption bands are saturated because there are thick original commercial samples. In Figure 7a,b, the changes in the hydroxyl domain for EVA sheets are significant and well appreciable, while, in the carbonyl domain, the presence of only a small shoulder at ca. 1780 cm^−1^ can be observed. Similarly, for POE sheets, the changes in the hydroxyl domain are noticeable, while, in the carbonyl domain a small shoulder at ca. 1640 cm^−1^, due to presence of insaturations, is barely noticeable.

In Figure 8a,b, the variations of the total band areas in hydroxyl domains for EVA and POE are plotted, respectively. Worth noting that the EVApre-lam and POEpre-lam samples show larger hydroxyl accumulations than the EVApost-lam and POEpost-lam samples, especially at long exposure time. Moreover, in Figure 8c,d, the pre-laminated samples show more pronounced increases for the shoulders at 1780 cm^−1^ for EVA and at 1640 cm^−1^ for POE, rather than the post-laminated samples. All these results highlight that the lamination process has a beneficial effect also on photoxidation resistance and again, it seems that the POE show better photoxidation resistance than the EVA one.

Further confirmation comes also by ATR-FTIR analysis of the investigated samples, in Figure 9a–d, the ATR-FTIR spectra of EVApre-lam, EVApost-lam, POEpre-lam and POEpost-lam commercial samples, before exposure and at maximum UVB exposure time are plotted. Therefore, to confirm the presence of some chemical species, the ATR-FTIR analysis can be considered suitable for qualitative surface analysis. The bands in both hydroxyl and carbonyl domains in the spectra of the four investigated samples at maximum exposure time appear larger than the same bands before exposure, and the latter is clearly exacerbated for the pre-laminated samples, confirming again the beneficial effect of the lamination process on the photoxidation resistance.

### 3.7. Thermo-Oxidation Resistance

In Figure 10a,b, FTIR spectra of EVApost-lam and POEpost-lam as a function of thermo-oxidation time are plotted, respectively.

The monitoring of thermo-oxidation process was extended up to ca. 3500 h because no significant variations were noticeable prior. It is worth noting that EVApost-lam sample shows a slight increase of the absorption band in the hydroxyl range and there is the appearance of a small shoulder at 1780 cm^−1^, suggesting the occurrence of oxidation phenomenon, see Figure 10a. Interestingly, the POEpost-lam sample is extremely stable up to ca. 3500 h thermo-oxidation, i.e., no significant variations in carbonyl and hydroxyl range are noticeable, highlighting no noteworthy occurrence of the oxidation phenomenon also at this prolonged thermal treatment, see Figure 10b. Considering this qualitative analysis, it can be summarized that the POEpost-lam sample is more stable and resistant to thermo-oxidation occurrence than EVApost-lam.

## 4. Conclusions

Accurate characterization of pre- and post-laminated EVA and POE industrial sheets was carried out by calorimetric, rheological and mechanical analysis. Obtained results suggest that the lamination process has a beneficial effect on the 3D-structuration on both polymers, though it seems that better results are obtained for POE sheets. Upon lamination process, in the melt state, the viscosity of POE increased, while the viscosity of EVA decreased. The latter could be understood considering that the EVA experiences degradation at high temperature with the formation of volatile acetic acids.

The durability, in terms of photo- and thermo-oxidation resistance, of EVA and POE sheets is evaluated monitoring the formation of new oxygen-containing species with absorption bands in the hydroxyl and carbonyl domain. The lamination process leads to the formation of more oxygen resistant sheets, and this is exacerbated for the POE sample.

Finally, to sum up, although both EVA and POE sheets can be considered suitable for encapsulant for bifacial heterojunction PV modules, the POEpost-lam sheet is better structured in the melt, it has good rigidity and ductility and is more stable, in terms of photo- and thermo-oxidation, than EVApost-lam.

## Figures and Tables

**Figure 1 polymers-14-01052-f001:**
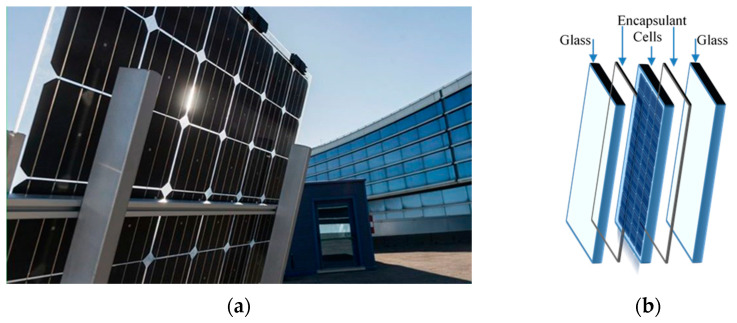
(**a**) New novel high reliable bifacial heterojunction glass/glass PV module and (**b**) its schematic structure (by 3SUN-ENEL Green Power. Catania, Italy).

**Figure 2 polymers-14-01052-f002:**
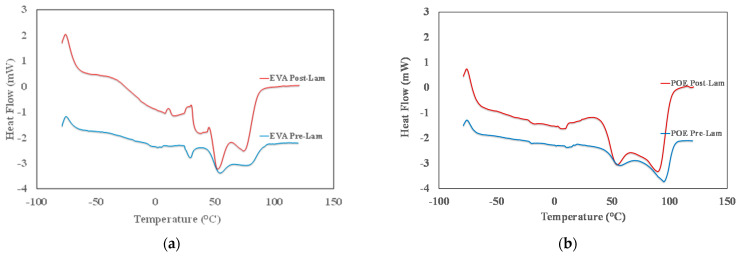
DSC thermograms of (**a**) EVApre-lam and EVApost-lam and (**b**) POEpre-lam and POEpost-lam commercial samples.

**Figure 3 polymers-14-01052-f003:**
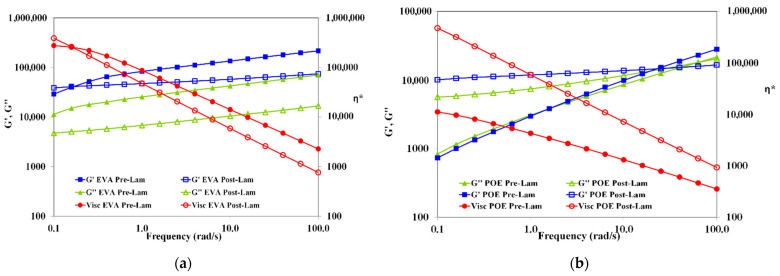
Storage and loss moduli (G’ and G”) and complex viscosity (η*) of (**a**) EVApre-lam (full symbols) and EVApost-lam (open symbols) and (**b**) POEpre-lam (full symbols) and POEpost-lam (open symbols) commercial samples.

**Figure 4 polymers-14-01052-f004:**
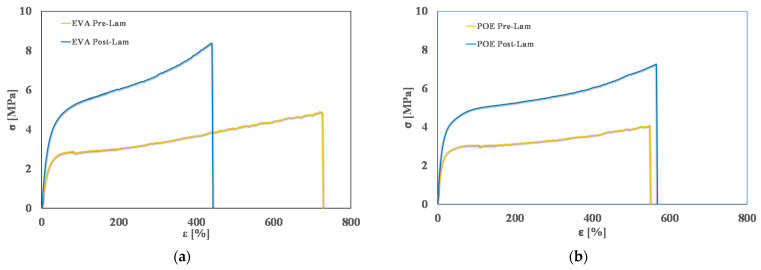
Stress-strain curves of (**a**) EVApre-lam and EVApost-lam and (**b**) POEpre-lam and POEpost-lam commercial samples.

**Figure 5 polymers-14-01052-f005:**
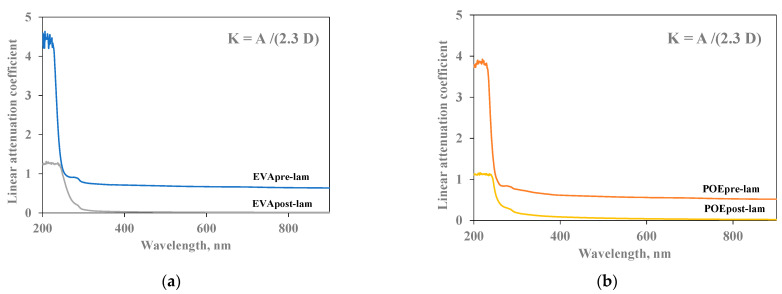
Linear attenuation coefficient (K) of (**a**) EVApre-lam and EVApost-lam and (**b**) POEpre-lam and POEpost-lam commercial samples.

**Figure 6 polymers-14-01052-f006:**
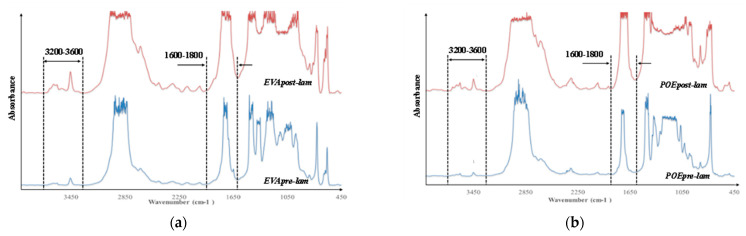
FTIR spectra of (**a**) EVApre-lam and EVApost-lam and (**b**) POEpre-lam and POEpost-lam commercial samples.

**Figure 7 polymers-14-01052-f007:**
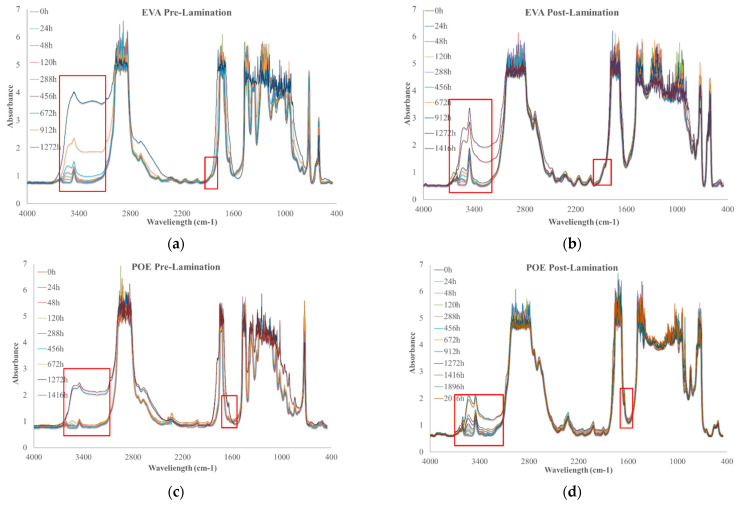
FTIR spectra of (**a**) EVApre-lam, (**b**) EVApost-lam, (**c**) POEpre-lam and (**d**) POEpost-lam commercial samples as a function of photo-oxidation time.

**Figure 8 polymers-14-01052-f008:**
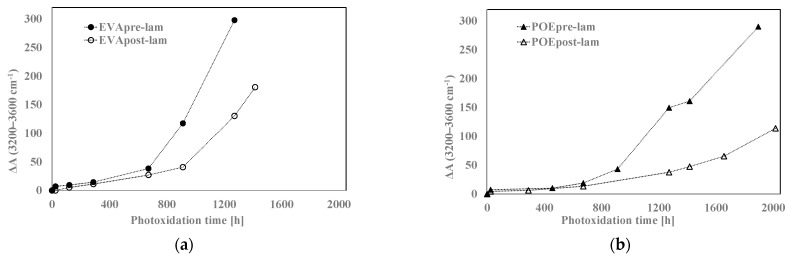
Variation of absorbance (**a**,**b**) at 3200–3600 cm^−1^ for EVA and POE samples, respectively, (**c**) at 1780 cm^−1^ for EVA and (**d**) at 1640 cm^−1^ for POE.

**Figure 9 polymers-14-01052-f009:**
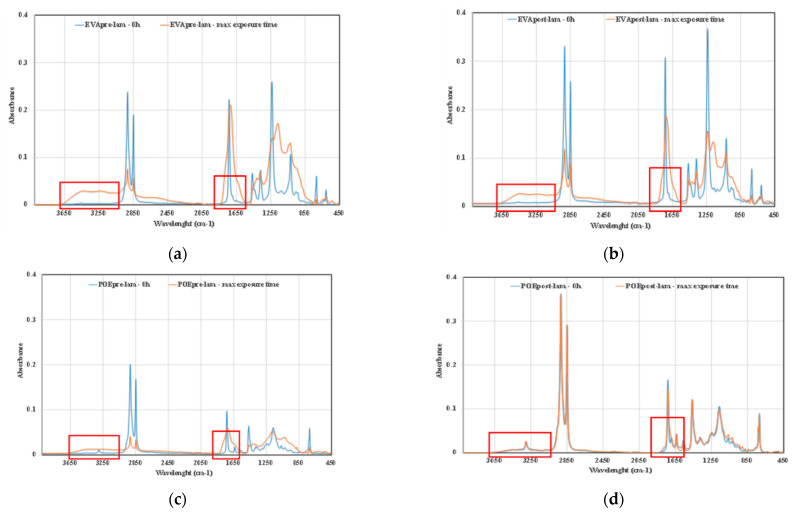
ATR-FTIR spectra of (**a**) EVApre-lam, (**b**) EVApost-lam, (**c**) POEpre-lam and (**d**) POEpost-lam commercial samples, before exposure and at maximum UVB exposure time.

**Figure 10 polymers-14-01052-f010:**
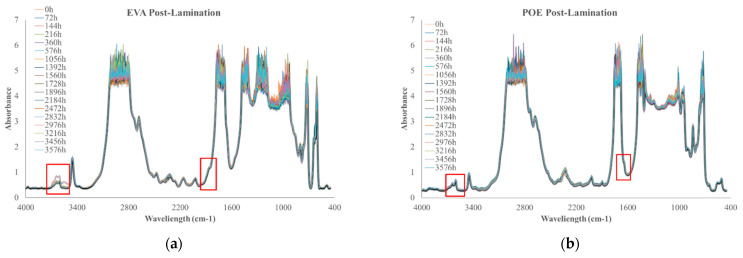
FTIR spectra of (**a**) EVApost-lam and (**b**) POEpost-lam as a function of thermo-oxidation time.

**Table 1 polymers-14-01052-t001:** Glass transition (Tg), exothermic and endothermic peaks (Tc and Tf) and fusion enthalpy (ΔH) of pre- and post-laminated EVA and POE samples.

	Glass Transition	Exothermic Phenomenon	Endothermic Phenomenon			
	Tg, °C	Tc, °C	Tf1, °C	Tf2, °C	Tf3, °C	ΔH, J/g
EVApre-lam	−66.9	5.27	29.92	54.78	78.77	12.32
EVApost-lam	−63.0	8.15/25.40 (*)	37.32	51.45	73.16	32.13
POEpre-lam	−67.4	12.02	29.26	58.36	78.11	12.54
POEpost-lam	−62.6	7.06	37.66	55.03	74.49	46.83

Note: (*) this exothermic peak appears as a complex peak and the temperature identification of is difficult.

**Table 2 polymers-14-01052-t002:** Main mechanical properties, i.e., elastic modulus (E), tensile strength (TS) and elongation at break (EB) of pre-laminated and post-laminated EVA and POE samples.

	E, MPa	TS, MPa	EB, %
EVApre-lam	11.6 ± 0.7	4.9 ± 0.3	725 ± 45
EVApost-lam	16.5 ± 1.2	8.3 ± 0.5	441 ± 27
POEpre-lam	21.4 ± 1.5	4.9 ± 0.3	550 ± 25
POEpost-lam	31.6 ± 2.5	7.3 ± 0.5	567 ± 25

## Data Availability

Not applicable.

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
