# Peer review of "Durability and Performance of Encapsulant Films for Bifacial Heterojunction Photovoltaic Modules"

_polymers, 2022, doi:10.3390/polym14051052_

Round 1
Reviewer 1 Report
The authors characterized EVA and POE materials as encapsulants of solar cells. The results are well summarized, and it was shown that POE is a better choice over EVA. The results will be a good reference in the solar cell manufacturing industry.
Some minor English corrections are recommended. For example, line 18 on page 1: To carried out => To carry out
Author Response
Thank you for your very positive evaluation of the paper.
Some English corrections have been made in revised manuscript.
Reviewer 2 Report
The authors showed POE sheets as more stable and more suitable candidate as encapsulating material for photovoltaic modules compared to EVA sheets. The authors conducted multiple material characterization experiments to support the conclusion, including dsc, FTIR, UV-vis, photo and thermal oxidation resistance experiments. The reviewer recommends this work to be published in Polymers given the following points to be addressed:
- Baiamonte et al. (ref 20) has demonstrated that the presence of polyolefin content in PVA blends enhances the properties and performances of encapsulants for bifacial heterojunction PV modules compared to neat PVA. Therefore, it is not surprising that the neat PVA was outperformed by POE blends in terms of the mechanical and stability properties heterojunction technology for PV modules (this study). The reviewer believes that the novelty of this study can be improved if the motivation of this study can be further explained.
- Although the reviewer understands that both POE and PVA sheets were purchased from commercial sources, the reviewer is concerned about the amount of crosslinking and stabilizing additives that were added to these sheets. Is there any difference in the amount of additives added to the POE, PVA sheets, respectively? Since these are "stabilizing" additives, they should have an impact on the stability properties of the encapsulants. The authors should explain or perform control experiments to address this.
- For FTIR experiments, the author described "In the spectra of EVApost-lam, the shoulder at ca. 1650 cm-1 274 is not visible, also because the carbonyl bands appear larger due to higher sample thick- 275 ness, while a small shoulder at ca. 1780 cm-1 appears, probably, due to the formation of 276 some esters during prolonged lamination process. " The reviewer is curious about the formation of ester during the lamination process. Perhaps some further explanation or citation to similar studies might help potential readers to understand the phenomenon.
Author Response
Thank you for your positive evaluation of the paper. Following you find authors’ comments about your questions:
- In our previous work, ref. 20 (updated now), we tried to formulate EVA-based encapsulants containing polyolefin because it has a beneficial effect on thermal resistance at high “hot-spot” temperature that could be randomly achieved for heterojunction PV modules. According to literature, the random “hot-spot” temperature increase is the main cause for the failure of PV-modules.
Therefore, as suggested by reviewer, the authors added further motivation for the current work. Please, see the text in green colour.
- The authors reported in the experimental part the kinds of additives (i.e. crosslinking agent, antioxidants and hindered amine light stabilizers), but the information about their amounts and chemical formulas are not available by producer.
The authors think that this work gives an important contribution to the state-of-the-art, suggesting that the EVA is a very good encapsulant material, but it is not excellent, as it prompts. The low cost of EVA makes it very attractive for the PV module producers, but exploring for other suitable polymer materials is a challenging issue.
- Regarding the esters formation at very low amount during the lamination, the authors think that this could be related to the progress of EVA crosslinking, and it depends on chemical nature/structure of used peroxide. This is an authors’ opinion, and it needs to be demonstrated before ascertaining in the scientific literature. However, this issue is out of scope of current work and for this reason, the authors prefer not to add any comment.
This manuscript is a resubmission of an earlier submission. The following is a list of the peer review reports and author responses from that submission.
Round 1
Reviewer 1 Report
I cannot read the figure data. The font size is too small, and image resolution is too low.
The authors must resubmit it with high quality figures.
Reviewer 2 Report
Page 1, lines 37-38: Please check this sentence, something is missing
Page 2, lines 54-55: You write: “…EVA degrades rapidly upon solar exposure, even if using appropriate crosslinking agents and stabilizers [12-14]….” As a general statement this is not true. There are EVA types that are very stable, not showing any signs of degradation due to sufficient stabilization, and then there are EVA types with insufficient stabilization showing degradation effects like yellowing.
Page 3, line 111: Superscript is missing at cm-1, check the whole document
Section 3.2 I believe the temperatures of the isotherm rheological measurements have been to high, so you don´t see much difference between the crosslinked and not crosslinked state. See Figure 8 in https://doi.org/10.1016/j.solmat.2013.04.022 , there you clearly see that at 170°C the measured shear viscosity is not influenced by the curing anymore. 20 minutes of lamination does not cause EVA degradation
Section 3.3 The increase in stress levels can be attributed to reduced mobility of the crosslinked polymer chains. So more energy is needed to deform the polymer, see Figure 4 in https://doi.org/10.1016/j.solmat.2013.04.022
Section 3.6 and 3.7 The practical relevance of these results are limited. First, the degradation reactions of EVA and other encapsulants are well described in literature. Moreover, the UV313 lamps you used provide much more low wavelength UV than the sun. So the results are not easily comparable to aging under natural sunlight, as maybe different degradations mechanisms are triggered by the different spectrum. Also, usually the materials are not directly exposed to sunlight, but encapsulated between glass. So the microclimate is different and definitely leads to different degradation reactions See https://doi.org/10.1016/j.rser.2017.06.039